# Functionalized TiO_2_ Nanotube-Based Electrochemical Biosensor for Rapid Detection of SARS-CoV-2

**DOI:** 10.3390/s20205871

**Published:** 2020-10-17

**Authors:** Bhaskar S. Vadlamani, Timsy Uppal, Subhash C. Verma, Mano Misra

**Affiliations:** 1Chemical and Materials Engineering Department, University of Nevada, Reno, NV 89557, USA; bvadlamani@unr.edu; 2Department of Microbiology and Immunology, University of Nevada, Reno School of Medicine, Reno, NV 89557, USA; tuppal@med.unr.edu

**Keywords:** SARS-CoV-2, COVID-19, diagnostics, rapid detection, spike glycoprotein, TiO_2_, metal oxide

## Abstract

The COronaVIrus Disease (COVID-19) is a newly emerging viral disease caused by the severe acute respiratory syndrome coronavirus 2 (SARS-CoV-2). Rapid increase in the number of COVID-19 cases worldwide led the WHO to declare a pandemic within a few months after the first case of infection. Due to the lack of a prophylactic measure to control the virus infection and spread, early diagnosis and quarantining of infected as well as the asymptomatic individuals are necessary for the containment of this pandemic. However, the current methods for SARS-CoV-2 diagnosis are expensive and time consuming, although some promising and inexpensive technologies are becoming available for emergency use. In this work, we report the synthesis of a cheap, yet highly sensitive, cobalt-functionalized TiO_2_ nanotubes (Co-TNTs)-based electrochemical sensor for rapid detection of SARS-CoV-2 through sensing the spike (receptor binding domain (RBD)) present on the surface of the virus. A simple, low-cost, and one-step electrochemical anodization route was used for synthesizing TNTs, followed by an incipient wetting method for cobalt functionalization of the TNTs platform, which was connected to a potentiostat for data collection. This sensor specifically detected the S-RBD protein of SARS-CoV-2 even at very low concentration (range of 14 to 1400 nM (nano molar)). Additionally, our sensor showed a linear response in the detection of viral protein over the concentration range. Thus, our Co-TNT sensor is highly effective in detecting SARS-CoV-2 S-RBD protein in approximately 30 s, which can be explored for developing a point of care diagnostics for rapid detection of SARS-CoV-2 in nasal secretions and saliva samples.

## 1. Introduction

The current outbreak of novel coronavirus (nCoV-2019 or SARS-CoV-2), was first detected in Wuhan, China in December 2019, but quickly spread to other parts of China as well as to the entire world, causing a pandemic [1]. According to the WHO, as of 16 August, 2020, around 21,294,845 people are infected, and 761,779 people have died due to SARS-CoV-2 infection [2]. SARS-CoV-2 infection causes a variety of symptoms including fever, cough and respiratory distress, which are collectively called the coronavirus disease or COVID-19 [3]. The spreading of SARS-CoV-2 primarily occurs from person-to-person transmission through close contact or via small droplets produced during coughing, sneezing, and talking [4,5]. The incubation period for SARS-CoV-2 is around 2–7 days, with no noticeable symptoms; however, the viral transmission from an infected person to a non-infected person is still possible during this asymptomatic period [6]. Under the current scenario, with no vaccines in the market, global lockdown regulations are in place in order to minimize the viral spread. Consequently, this pandemic has caused a severe socio-economic impact on the world economy and raised fears of a global recession [7]. Currently, the real-time reverse-transcriptase polymerase chain reaction (RT-PCR) technique is the most common and reliable laboratory testing method for qualitative/quantitative SARS-CoV-2 detection [8,9] followed by serum virus neutralization assay (SVNA) for the determination of antibody neutralization [10] and enzyme-linked immunoassays (ELISA) for the detection of antibodies against SARS-CoV-2 [11]. However, the major limitations of these laboratory-based diagnostic tests are the invasive nature of the tests that often require trained personal for nasopharyngeal sample collection, along with the requirement of highly sophisticated machines, cross-reactivity with other viruses, and longer duration of testing. In order to contain the viral spread, surveillance of even asymptomatic individuals is needed, which is feasible only after the development of a simple, portable and rapid point-of-use sensor for the detection of SARS-CoV-2.

SARS-CoV-2 has a positive-sense, single-stranded RNA (~30K bp) genome with 14 ORFs that encode for structural, replication and non-structural proteins [12]. Similar to its genetic cousin, human SARS-CoV, SARS-CoV-2 consists of four structural proteins viz. spike (S), envelope (E), membrane (M), and nucleocapsid (N). Coronaviruses are named for the crown like spike glycoprotein, S (composed of two subunits: the S1 subunit and the S2 subunit) on the surface/envelope [13]. The S1 subunit of the S protein consists of a receptor binding domain (RBD) that has a high binding affinity towards the host angiotensin-converting enzyme II (ACE2) receptor present on the human cells; the S2 subunit mediates virus-host cell fusion and entry [14]. Importantly, the S protein is highly immunogenic and induces immune response to produce neutralizing antibodies as well as T-cell responses in SARS-CoV-2 infected individuals [15]. Functionally, binding of S-RBD to the hACE2 receptor is crucial for the entry of SARS-CoV-2 into human cells. Interestingly, SARS-CoV-2 S-RBD shares only 70% sequence identity with SARS-CoV S-RBD, which has been evaluated for vaccines and therapeutic drug development [16]. Hence, the S-RBD of SARS-CoV-2 is an excellent target for diagnostic and therapeutic interventions.

Electrochemical biosensors are advantageous for sensing biomolecules because of their ability to detect biomarkers with accuracy, specificity and high sensitivity [17]. Electrochemical biosensors have been successfully used in medical diagnostics for the detection of viruses such as the Middle East respiratory syndrome coronavirus (MERS-CoV) [18], the human enterovirus 71 (EV71) [19], the human influenza A virus H9N2 [20], and the avian influenza virus (AIV) H5N1 [21]. Lahyquah et al. [18] used an array of carbon electrodes modified with gold nanoparticles for the detection of MERS-CoV. Very recently, a biosensor using gold nanoparticle decorated FTO glass immobilized with nCovid-19 monoclonal antibody was reported for the detection of SARS-CoV-2 [22]. The functionality of the electrochemical biosensor can be further improved by nanostructuring the electrode as it increases the electrochemical reaction rate due to an increased electrode surface area to volume ratio, thereby increasing the electrode surface area to analyte fluid volume. In the work by Chin et al. on the encephalitis virus, it was found that nanostructuring of carbon electrodes with carbon nanoparticles increased the current response by 63% due to an enhanced electron charge transfer kinetics [23]. Similarly, we have reported that Co functionalized TiO_2_ nanotubes (Ni-TNTs) with higher surface-to-volume ratio can detect the biomarkers associated with tuberculosis [24,25]. The proposed sensing mechanism involves the formation of a complex between Co and the biomarker at a specific bias voltage, due to the reduction of Co ions and oxidation of the biomarker. Similarly, we hypothesized that S-RBD or SARS-CoV-2 can be detected through complexing of functionalized nanoparticles with the S-RBD protein. A schematic of viral detection directly from a patient sample is shown in Figure 1.

In the current work, we have determined the potential of Co-functionalized TiO_2_ nanotubes (Co-TNTs) for the electrochemical detection of S-RBD protein of SARS-CoV-2. TNTs were synthesized by a simple, cost-effective, one-step electrochemical anodization route, and Co functionalization was carried out using the incipient wetting method. Our data shows that cobalt functionalized TNTs can selectively detect the S-RBD protein of SARS-CoV-2 using the amperometry electrochemical technique in ~30 s.

## 2. Materials and Methods

### 2.1. Synthesis of TNTs

TNTs were synthesized by electrochemical anodization of the Ti sheet. A Ti sheet of size 1.5 × 1.5 cm, with a tab 2 mm in width, was cut out of a G1 grade Ti sheet (thickness 0.01016 mm). One side of the coupon was polished with 600 grit polishing paper for 4 min to remove any surface metal oxide layer. The coupon was ultrasonicated in a 1:1 solution of ethanol and acetone for 2 min. The unpolished side was masked with Kapton tape to avoid any exposure to electrolyte during anodization. The electrochemical anodization was performed in a standard two-electrode configuration, using Ti foil as a working electrode and platinum foil as a counter electrode with a 3 cm gap between them. The anodization was carried out using an electrolyte of composition 96.5 mL (CH_2_OH)_2_, 3 mL DI H_2_O, and 0.505 g NH_4_F, in a Teflon beaker. The electrolyte was maintained at a subzero temperature and was continuously stirred using a magnetic stirrer at a speed of 140 rpm. The anodization was carried out by maintaining a constant voltage of 30 V across both the electrodes for 50 min. After anodization, the sample was rinsed in DI H_2_O and baked in an oven at 120 °C for 4 h. The Kapton tape was removed from the sample after baking, and the sample was annealed in a tube furnace at 500 °C for 3 h in a continuous flow of oxygen.

### 2.2. Synthesis of Co functionalized TNTs

The annealed TNTs obtained from the furnace were functionalized with cobalt using an incipient wetting method, i.e., a wet ion exchange process. The same side of the sample that was masked earlier was again masked with Kapton tape. The sample was ultrasonicated in a solution containing 2.306 g of CoCl_2_.6H_2_O in 20 mL ethanol for 35 min. The sample was baked in an oven at 120 °C for 4 h to obtain cobalt functionalized TNTs.

### 2.3. SEM Characterization

The morphology of the TNTs and Co-TNTs were examined using dual beam scanning electron microscopy (SEM, ThermoFisher Scientific). The cobalt content in the Co-TNT sample was analyzed using the EDS detector attached to SEM. The SEM micrographs were analyzed using ImageJ software.

### 2.4. Synthesis and Purification of SARS-CoV-2 S-RBD Protein

The pCAGGS vector containing SARS-CoV-2 Wuhan-Hu-1 spike glycoprotein receptor binding domain (RBD) with a C-terminal hexa-histidine tag was obtained from BEI Resources (NIAID, NIH, NR-52309). His_6_-tagged S-RBD containing pCAGGS plasmid was expressed in HEK293T (human embryonic kidney) cells obtained from the American type culture collection (ATCC) and maintained in Dulbecco’s modified Eagle medium (DMEM), supplemented with 10% fetal bovine serum (FBS, Atlanta Biologicals), 2 mM L-glutamine, 25 U/mL penicillin, and 25 μg/mL streptomycin. Cells were grown at 37 °C in a humidified chamber supplemented with 5% CO_2_. For His_6_-tagged S-RBD protein generation, HEK293T cells were transfected with recombinant plasmid using Neon Transfection system (Thermo Scientific) according to the manufacturer’s instructions. Supernatants from transfected cells were harvested on day 3 post-transfection and the cell debris was removed by centrifugation (4,000 rpm, 20 min at 4° C). Supernatants were then incubated with 1 mL of Ni-NTA Agarose (Qiagen) for every 10 mL of supernatant, for 2 h at 4 °C with rotation. For S-RBD purification, gravity flow columns were used to load the NI-NTA agarose bound His_6_-tagged spike-RBD protein, followed by washing with wash buffer (20mM sodium phosphate, 500 mM NaCl, 8 M urea, 20 mM imidazole, pH 6.0) and eluting with elution buffer (20 mM sodium phosphate, 500 mM NaCl, 8M urea, 200 mM imidazole, pH 4.0). The eluted protein was concentrated using protein concentrators (Thermo Scientific, 87,748 and 87772), quantified using Bradford assay and Nanodrop (Thermo Scientific) and further analyzed by SDS-PAGE.

### 2.5. Electrochemical Characterization

The electrochemical sensing of S-RBD protein was carried out using a custom-built Co-TNT packaged printed circuit board setup. The sensor response was measured with the help of Gamry reference 600+ potentiostat attached to the printed circuit board. The custom-made printed circuit board consists of a copper clamp that holds the Co-TNT grown over the Ti sheet. The upward-facing Co-TNT side acts as a working electrode, and the bottom-facing Ti side acts as a counter electrode, to which electrical connections were made via copper lines running on the top and bottom of the custom-built chip, respectively. The detailed schematic of the printed circuit board was reported in our earlier work [26]. The schematic of the whole sensing set up along with the detection methodology is shown in Figure 1. Amperometry is an electrochemical technique where a constant voltage is applied across the electrodes and response current is monitored as a function of time [27]. The technique uses response current to determine the concentration of the analyte in the electrolyte solution between the electrodes. The S-RBD protein in the elution buffer (20 mM sodium phosphate, 500 mM NaCl, 8 M urea, 200 mM imidazole, pH 4.0) was transferred onto the surface of Co-TNT using a micropipette. The sensor response with various S-RBD protein in concentrations was determined using the amperometry technique, at a bias voltage of −0.8 V. The bias voltage was determined by conducting the cyclic voltammetry experiments in the voltage window −2 to +2 V. All the experiments were carried out at room temperature.

## 3. Results and Discussion

### 3.1. Co-TNT Showed Characteristics Nanotube Formation

The scanning electron microscopy (SEM) micrographs of the TNTs, prepared by electrochemical anodization, are shown in Figure 2a. The inset shows the side view of the TNTs (Figure 2a). The outer diameter and wall thickness of TNTs were ~60 and ~10 nm, respectively. The average length of TNTs was found to be ~1.1 µm. In our earlier work, TNTs synthesized under similar conditions were found to show the crystalline anatase phase predominantly [25]. The surface morphology of the Co-TNTs examined under SEM is shown in Figure 2b. The SEM micrograph reveals the presence of precipitates on top of the TNT surface. EDS analysis confirmed the uniform distribution of Co on top of TNTs (Figure 2c), and the Co content was found to be ~4 wt% (Figure 2d). We have previously shown using detailed XPS studies that Co exists in the Co^+2^ state (2p_3/2_ peak at 781.5 eV) and also Co(OH)_2_ is the predominant phase present on the surface of Co-TNTs [28]. Therefore, the morphology of TNTs can be visualized as having a very large surface area, uniformly decorated with Co^+2^ ions.

### 3.2. S-RBD Protein Showed Specific Monomeric and Dimeric Forms

The receptor binding domain of the spike glycoprotein (S-RBD), present as a crown on the surface of the virus is an easily accessible target for the detection of SARS-CoV-2. The RBD domain comprises of amino acids 329-521, which is a ~25 kDa protein with potential N-glycosylation sites. As shown in Figure 3A, B, the SDS-PAGE gel of His_6_-tagged S-RBD protein, either stained with SimplyBlue SafeStain (Figure 3A) or immunologically detected with mouse anti-His monoclonal antibody (Figure 3B) showed the presence of specific protein in our viral protein preparation. Immunoblot detected SARS-CoV-2 S-RBD protein at approximately 35 kDa, as expected, but also at 70 kDa, representing the dimeric forms of S-RBD protein (Figure 3B). Detection of a slightly higher molecular weight (~35kda) S-RBD protein as compared to the calculated size was possibly because of post-translational modifications, including glycosylation on the protein. Importantly, the S-RBD purified protein from the human embryonic kidney cells were of high purity; it was used for quantitation and detection on Co-TNT sensors.

### 3.3. S-RBD Protein Was Detected on Co-TNTs Sensors

The ability of Co-TNT to sense the S-RBD protein of SARS-CoV-2 was determined by performing an amperometry experiment at a bias voltage of −0.8 V. The amperometry curves obtained at various concentrations of protein are shown in Figure 4. The sensor was exposed to protein 30 s after the beginning of the experiment (marked by an arrow). The sensor response current increases sharply and rapidly as the sensor was exposed to the protein. At a protein concentration of 1400 nM (nano molar), the peak sensor current output was found to be ~0.74 µA (micro ampere). The peak current decreases to ~0.45 µA at a protein concentration of 140 nM and further decreases to ~0.23 µA at a protein concentration of 14 nM. The sensor detection time was ~30 s over the concentration range of 14 to 1400 nM. It is hypothesized that the rapid increase in sensor response current could be attributed to the electrochemically triggered unfolding of protein that exposes its interior [29,30,31] and subsequent complex formation between Co and the protein [32,33]. Each S-RBD protein monomeric unit contains 15 Tyrosine, 2 Tryptophan and 9 Cysteine amino acid residues [14], and all of them were reported to underdo electrochemical oxidation under application of potential [29,32]. The electrochemical oxidation process involves deprotonation, where the -OH functional group in the protein is converted to -O^−^. We envisage that the complex formation occurs between the Co^+2^ ion in Co-TNT and the -O^−^ radical in the protein. A very similar mechanism was reported earlier, where methyl nicotinate biomarker was exposed to Co-TNT [28]. 

The average sensor response time, which is defined as the time taken to reach the peak current, was found to be ~2 s. It is very short compared to our earlier studies on the sensor for colorectal cancer, where a sensor response time of ~200 s was documented [33]. The shorter sensor response time indicates higher kinetics of the reaction between Co-TNT and the protein molecules. At all the protein concentrations, it was observed that the sensor current did not recover to the initial baseline current within the experimental timeframe. Therefore, the sensor recovery time, defined as the sensor’s time to recover to the initial baseline current value, could not be reported. The sensor’s strange behavior could be due to the change in the surface chemistry of Co-TNT after interaction with the protein.

### 3.4. Sensor Response Measurement

The sensor response (SR) was calculated at various protein concentrations based on the following equation:$$\mathrm{Sensor}\;\mathrm{Resposne}\;\left(\mathrm{SR}\right)=\frac{i_{\max,\;\mathrm{protein}}-i_{\max,\;\mathrm{base}\;\mathrm{line}}}{i_{\max,\mathrm{base}\;\mathrm{line}}}$$
where $i_{\max,\mathrm{protein}}$ is the maximum current obtained when the sensor is exposed to SARS-CoV-2 S-RBD protein and $i_{\max,\mathrm{base}\;\mathrm{line}}$ is the maximum current obtained when the sensor is not exposed to the protein. The value of $i_{\max,\mathrm{base}\;\mathrm{line}}$, which is the current obtained when the sensor is not exposed to the protein, was found to be ~10 pA (Figure 4). The sensor responses measured at different protein concentrations are shown in Figure 5. The sensor response was found to increase with an increase in the concentration of protein. Moreover, the sensor response exhibited excellent linearity over the concentration range 14 to 1400 nM with a correlation coefficient of R^2^ = 0.99. The regressed linear calibration curve for sensor response was obtained as follows:$$\mathrm{SR}=\left(0.266\pm 0.026\right)\log\left(C\right)+\left(4.053\pm 0.059\right);\;R^{2}\;=\;0.99$$
where SR is the sensor response, and C is the concentration of protein in nM. According to statistical analysis [34], the detection limit fof measurements using the sensor was determined to be 0.7 nM.

The limit of detection can be further improved by the use of (i) Co-TNT synthesized by an in-situ anodization technique and (ii) Co-TNTs of even higher length. Previously, we found that Co-TNT synthesized by in-situ anodization with higher sensor sensitivity compared to Co-TNT synthesized by the incipient wetting route, towards the detection of tuberculosis biomarkers [25]. A higher sensor sensitivity corresponds to a better limit of detection and sensitivity of quantitation. The increased sensitivity was attributed to the presence of Co (OH)_2_ precipitate sites in direct contact with the parent TiO_2_, due to which direct conduction is possible. The sensor sensitivity can also be improved by using longer Co-TNTs as higher surface area results in a higher reaction rate; thereby, higher sensor response current can be obtained even at lower protein concentrations. 

## 4. Conclusions

In this study, we developed a Co-metal functionalized TNT as a sensing material for electrochemical detection of SARS-CoV-2 infection through the detection of the receptor binding domain (RBD) of spike glycoprotein. We confirmed the biosensor’s potential for clinical application by analyzing the RBD of the spike glycoprotein on our sensor. Amperometry electrochemical studies indicated that the sensor could detect the protein in the concentration range 14 to 1400 nM. The relationship between sensor response and protein concentration was found to be linear with the limit of detection as low as ~0.7 nM levels. Importantly, our sensor detected SARS COV-2 S-RBD protein in a very short time (~30 s), confirming its implication in developing a rapid diagnostic assay. Our report thereby demonstrates the development of a simple, inexpensive, rapid and non-invasive diagnostic platform that has the potential of detecting SARS-CoV-2 on clinical specimens, including nasal, nasopharyngeal swabs or saliva. Moreover, the developed approach has the potential for diagnosis of other respiratory viral diseases by identifying appropriate metallic elements to functionalize TNTs.

## Figures and Tables

**Figure 1 sensors-20-05871-f001:**
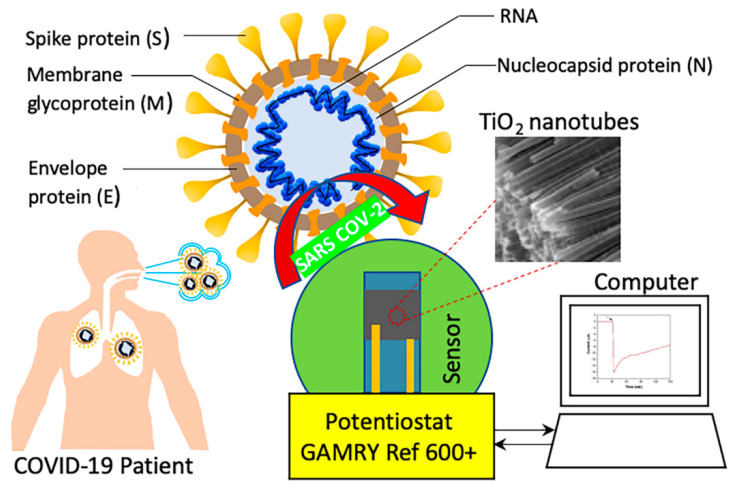
Schematic of Co-functionalized TiO_2_ nanotube (Co-TNT)-based sensing platform for the detection of SARS-CoV-2.

**Figure 2 sensors-20-05871-f002:**
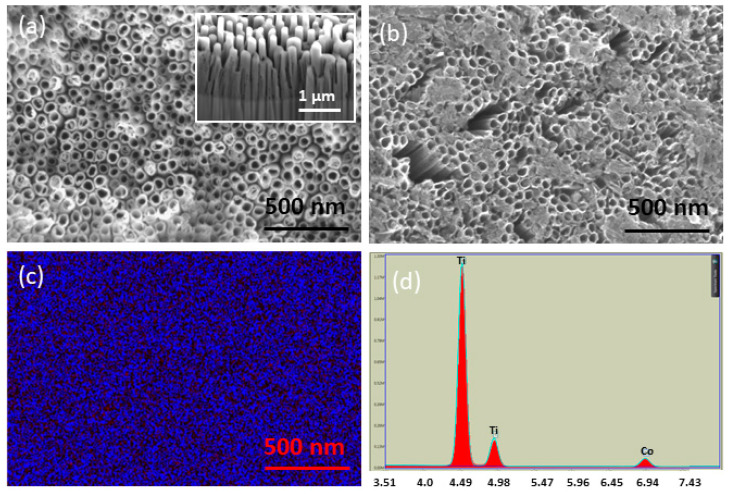
Scanning electron microscopy (SEM) micrographs of (**a**) TiO_2_ nanotubes (TNTs) post-annealing. Inset shows sidewalls of TNTs, (**b**) Co-functionalized TNTs showing the Co (OH)_2_ precipitate, (**c**) EDS map of Co confirming its uniform distribution, and (**d**) EDS spectra confirming the presence of Co.

**Figure 3 sensors-20-05871-f003:**
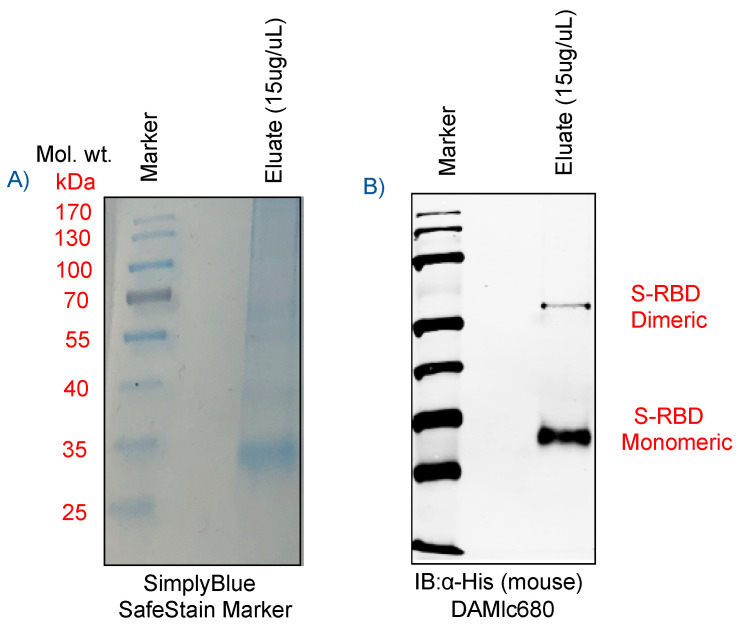
(**A**) Detection of purified receptor binding domain (RBD) protein of spike glycoprotein of SARS-CoV-2 stained with SimplyBlue dye, and (**B**) immunoblot to confirm the His_6_-tagged S-RBD protein (purified under denaturing conditions) probed with mouse anti-His monoclonal antibody. SARS-CoV-2 S-RBD characteristic bands were detected at 35 kDa/monomeric form and 70 kDa/dimeric form, respectively.

**Figure 4 sensors-20-05871-f004:**
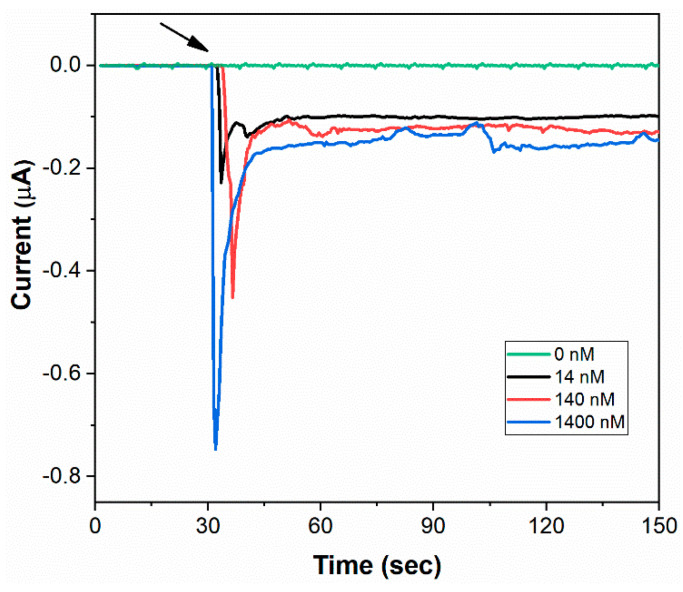
Amperometry response curves of Co-TNT sensor, at a bias voltage of −0.8 V, upon exposure to SARS-CoV-2 S-RBD protein of concentrations 0 (background), 14, 140, and 1400 nM.

**Figure 5 sensors-20-05871-f005:**
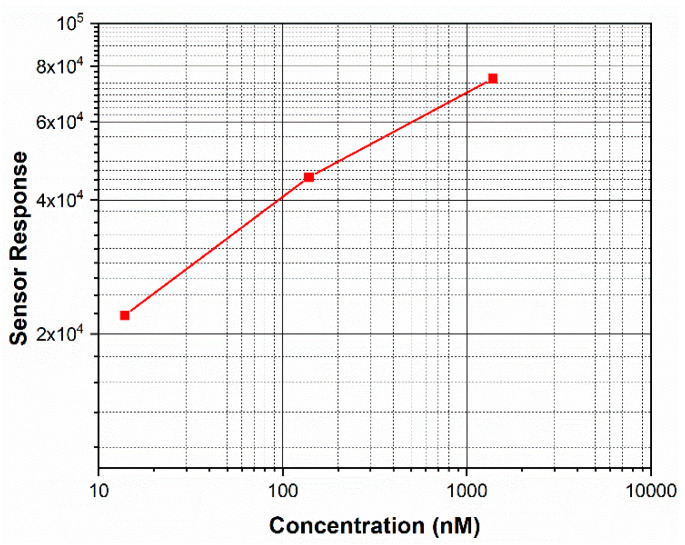
Plot showing the effect of SARS-CoV-2 S-RBD protein concentration on the variation of the sensor response for the Co-TNT sensor. The sensor response shows a linear region from 14 to 1400 nM.

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
