# Peer review of "Functionalized TiO2 Nanotube-Based Electrochemical Biosensor for Rapid Detection of SARS-CoV-2"

_sensors, 2020, doi:10.3390/s20205871_

Round 1
Reviewer 1 Report
In this manuscript, a functionalized TiO2 nanotube (Co-TNTs)-based electrochemical biosensor for rapid detection (in 30 seconds) of spike glycoprotein of SARS-CoV-2 by examining S-RBD protein. This study prefers to focus on a hot issuer about SARS-CoV-2 in this year. The functionalized TiO2 nanotube (Co-TNTs)-based electrochemical biosensor has been widely reported (such as https://doi.org/10.1016/j.apsusc.2018.11.032 ; https://doi.org/10.1016/j.bios.2016.02.004 ), but this method is not new and not innovative in biosensor filed.
Major changes: This manuscript need more detail data and information for electrochemical biosensor such as, XPS spectra, Table. comparisons of the various sensor responses, and so on (please also find related changes from reference: https://doi.org/10.1016/j.bios.2016.02.004 ; https://doi.org/10.1016/j.apsusc.2018.11.032 ) and need more detailed discussions, especially about Fig.3 and Fig.4.
Minor changes:
- Page 2: “…the major limitations of these laboratory based diagnostic tests is…”. Should be “are” not “is”.
- Page 2: “Infringingly, …”. The meaning is not clear.
- Page 6: “Using statistical analysis the limit of detection for measurements made using sensor was determined to be 0.7 nM.”
It should be written as: “according to statistical analysis, the detection limit of measurements using the sensor was determined to be 0.7 nM.”
- Page 7: “…due to which direct conduction is possible.”
The meaning of this sentence is not clear.
Author Response
In this manuscript, a functionalized TiO2 nanotube (Co-TNTs)-based electrochemical biosensor for rapid detection (in 30 seconds) of spike glycoprotein of SARS-CoV-2 by examining S-RBD protein. This study prefers to focus on a hot issuer about SARS-CoV-2 in this year. The functionalized TiO2 nanotube (Co-TNTs)-based electrochemical biosensor has been widely reported (such as https://doi.org/10.1016/j.apsusc.2018.11.032; https://doi.org/10.1016/j.bios.2016.02.004), but this method is not new and not innovative in biosensor filed.
RESPONSE: Thank you very much for your time and effort in reviewing our manuscript. We appreciate your salient observation on the need of a sensor for widespread screening of COVID-19 patient through the detection of either the viral material, cellular biomarkers or a combination of both for identifying individuals with SARS-CoV-2 infection. We agree that detection of various markers through TiO2 sensors has been previously used including us for the detection of TB biomarkers, which makes this technology to be easily deployable for a cost-effective detection of SARS-CoV-2 infection and screening of even among asymptomatic individuals. Here, we showed that TiO2 sensor can detect the surface protein (spike glycoprotein), which is present on the outermost layer of the virus.
Major changes: R1.1: This manuscript need more detail data and information for electrochemical biosensor such as, XPS spectra, Table. comparisons of the various sensor responses, and so on (please also find related changes from reference: https://doi.org/10.1016/j.bios.2016.02.004; https://doi.org/10.1016/j.apsusc.2018.11.032) and need more detailed discussions, especially about Fig.3 and Fig.4.
Response:
We agree and following changes are made in the revised manuscript. Authors have carried out very detailed XPS studies on Co-TNTs in past and results are published in our earlier work by Bhattacahrya et al [26]. To clearly reflect this in manuscript, we modified the existing sentence “We have previously shown that Co exists in Co+2 state in the form of Co(OH)2 on Co-TNTs [26]” to
“We have previously shown using detailed XPS studies that Co exists in Co+2 state (2p3/2 peak at 781.5 eV) and also Co(OH)2 is the predominant phase present on the surface of Co-TNTs [28].” (Line 177)
We added EDS mapping that conformed uniform distribution of Co (Figure 2c). Also, added EDS spectra that shows the presence of Co (Figure 2d). The sentence “EDS analysis confirmed the uniform distribution of Co on top of TNTs, and the Co content was found to be ~ 4 wt %.” was modified to “EDS analysis confirmed the uniform distribution of Co on top of TNTs (Figure 2c), and the Co content was found to be ~ 4 wt % (Figure 2d).” (Line 176)
We have also added additional information about Figure 3 and the text reads as:
The Receptor Binding Domain of the Spike glycoprotein (S-RBD), present as crown on the surface of the virus is an easily accessible target for the detection of SARS-CoV-2.
As shown in Figure 3a-b, the SDS-PAGE gel of His6-tagged S-RBD protein, either stained with SimplyBlue SafeStain (Fig. 3a) or immunologically detected with mouse anti-His monoclonal antibody (Figure 3b) showed the presence of specific protein in our viral protein preparation. Immunoblot detected SARS-CoV-2 S-RBD protein at approximately 35kDa, as expected but also at 70kDa, representing the dimeric forms of S-RBD protein (Fig. 3B). Detection of slightly higher molecular weight (~35kda) S-RBD protein as compared to the calculated size was possibly because of post-translational modifications, including glycosylation on the protein. Importantly, the S-RBD purified protein from the human embryonic kidney cells were of high purity, which was used for quantitation and detection on Co-TNT sensors. (Line 182)
Figure legend (Fig. 3) has been modified as well for clarity.
The following discussion was added regarding Figure 4 at the end of Section 3.3.
“At all the protein concentrations, it was observed that the sensor current did not recover to the initial baseline current within the experimental timeframe. Therefore, the sensor recovery time, defined as the sensor's time to recover to the initial baseline line current value, could not be reported. The sensor's strange behavior could be due to the change in the surface chemistry of Co-TNT after interaction with protein.” (Line 215)
Minor changes:
R1.2: Page 2: “…the major limitations of these laboratory based diagnostic tests is…”. Should be “are” not “is”.
Response:
We agree with the Reviewer#1. The grammatical correction has been corrected in the revised manuscript. (Line 54)
R1.3: Page 2: “Infringingly, …”. The meaning is not clear.
Response:
Thank you for the comment. The word “Infringingly” is typo and it was changed to “Interestingly” in the manuscript. (Line 70)
R1.4: Page 6: “Using statistical analysis the limit of detection for measurements made using sensor was determined to be 0.7 nM.” It should be written as: “according to statistical analysis, the detection limit of measurements using the sensor was determined to be 0.7 nM.”
Response:
We agree with your comment. As suggested, the sentence “Using statistical analysis the limit of detection for measurements made using sensor was determined to be 0.7 nM” is changed to “According to statistical analysis, the detection limit of measurements using the sensor was determined to be 0.7 nM”. The change was highlighted in yellow in the manuscript. (Line 238)
R1.5: Page 7: “…due to which direct conduction is possible. “The meaning of this sentence is not clear.
Response:
We have explained the meaning of the sentence as follows: In our earlier work, HRTEM studies indicated that Co-TNTs synthesized by in-situ anodization technique contains the Co(OH)2 precipitates embedded in the matrix of TiO2. As a result, the electron transport path is effective due to direct contact between the Co(OH)2 reaction site and TiO2 matrix. In contrast, electron transport path in Co-TNT synthesized by incipient wetting method is randomized due to the many grain and particle/particle boundaries. We observed that the sensor response with Co-TNT synthesized by in-situ anodization is far higher compared to that with Co-TNT synthesized by incipient wetting method. It was envisaged that the improved sensor response was due to the direct conduction that occurs between Co(OH)2 precipitate and TiO2 matrix. However, the use of the of Co-TNT synthesized by in-situ anodization is not the scope of current work and we plan to pursue it in future.

Reviewer 2 Report
Please see below for the specific comments,
- Abstract is longer than needed.
- For the following sentence please make it a simple sentence for broad range of scientists by using a simple terminology if it is possible. Also, Co-TiO2 can be defined as inexpensive.
“In this work, we report the synthesis of a cheap yet highly sensitive cobalt functionalized TiO2 nanotubes (Co-TNTs)-based electrochemical biosensor and its efficacy for rapid detection of spike glycoprotein of SARS-CoV-2 by examining S-RBD protein as the reference material.”
- Remove the word in summary from the abstract.
- I suggest the inclusion of the metal oxide related key word to the keywords section.
- What is the meaning of the following word ‘Infringingly’?
- Spike protein chemistry, such as polymeric structure of it, at least minimally showing the elemental and bonding composition of it, is needed to be included in the Figure 1 briefly and simply.
- Can you define the amperometry electrochemical technique?
- Is there any result of sensing against spike protein without cobalt functionalization?
- Can you please provide details for the exact nature of the cobalt functionalization such as is it surface coating of Co3O4 oxide on TiO2 surface or is there doping effect going on ? Hydoroxide state of the cobalt preserved after 120°C overnight annealing as well?
- Can you please describe how Co-TNT was deposited on custom-built Co-TNT packaged printed circuit board setup? Also electrochemical sensor oriented audience will be extremely interested in the current collector/electrodes and architecture of them as well? Please briefly provide the details.
- Figure 2, EDS map is not visible.
- Can you please include how the S-RBD protein of SARS-CoV-2 was transferred onto the sensor surface? Is the S-RBD protein of SARS-CoV-2 in solution? What kind of solution is that? What are the properties of it?
- nano molar (nM) should have given as an explicit form in the beginning of the paper.
- For the following sentence, is it known which part (elemental) of the spike protein take places/involves in that complex formation between Co and protein?
“It is hypothesized that the rapid increase in sensor response current could be attributed to the electrochemically triggered unfolding of protein that exposes its interior [27][28][29] and subsequent complex formation between Co and the protein [31][32]”
- Please clarify the following sentence “(ii) Co-TNTs of even higher length.”
- For the following sentence, the concentrations tested in the current work is similar concentration can be found in the human nasal liquids?
“Amperometry electrochemical studies indicated that the sensor could detect the protein in the concentration range 14 nM to 1400 nM.”
Author Response
Comments and Suggestions for Authors
Please see below for the specific comments,
- Abstract is longer than needed.
RESPONSE: We thank you and appreciate your time and effort in reviewing our manuscript. Although Abstract may not be required for letters but we think that providing abstract will give the readers a quick view of this work. Some of the letters published in SENSORS have an abstract, so it may be allowed however, if it’s required to remove, we would be happy to take it out. Thank you very much.
- For the following sentence please make it a simple sentence for broad range of scientists by using a simple terminology if it is possible. Also, Co-TiO2 can be defined as inexpensive.
“In this work, we report the synthesis of a cheap yet highly sensitive cobalt functionalized TiO2 nanotubes (Co-TNTs)-based electrochemical biosensor and its efficacy for rapid detection of spike glycoprotein of SARS-CoV-2 by examining S-RBD protein as the reference material.”
RESPONSE: We have changed the sentence to make it simpler.
In this work, we report the synthesis of a cheap, yet highly sensitive, cobalt-functionalized TiO2 nanotubes (Co-TNTs)-based electrochemical sensor for a rapid detection of SARS-CoV-2 through sensing the Spike (Receptor Binding Domain; RBD) present on the surface of the virus. (Line 22)
- Remove the word in summary from the abstract.
RESPONSE: We have changed ‘in summary’ to ‘Thus’ (Line 30)
- I suggest the inclusion of the metal oxide related key word to the keywords section.
RESPONSE: We agree and have included ‘metal oxide’ as key word in the revised manuscript. (Line 33)
- What is the meaning of the following word ‘Infringingly’?
RESPONSE: We thank the Reviewer and have changed “Infringingly” typo to “Interestingly” in the revised manuscript. (Line 70)
- Spike protein chemistry, such as polymeric structure of it, at least minimally showing the elemental and bonding composition of it, is needed to be included in the Figure 1 briefly and simply.
RESPONSE: The protein has a folded structure. The polymeric structure is found in its interior and cannot be found outside. We therefore are not including it in the Figure 1.
- Can you define the amperometry electrochemical technique?
RESPONSE: We thank the Reviewer for the comment. The following sentence is added to the ‘Electrochemical Characterization’ section in the revised manuscript.
Amperometry is an electrochemical technique where a constant voltage is applied across the electrodes and response current is monitored as function of time. The technique uses response current to determine the concentration of the analyte in the electrolyte solution between the electrodes. (Line 158)
- Is there any result of sensing against spike protein without cobalt functionalization?
RESPONSE: We thank you for this excellent point. We found that TiO2 nanotubes without Co functionalization did not shown any sensor response with spike protein.
- Can you please provide details for the exact nature of the cobalt functionalization such as is it surface coating of Co3O4oxide on TiO2 surface or is there doping effect going on? Hydoroxide state of the cobalt preserved after 120°C overnight annealing as well?
RESPONSE: We thank the Reviewer for the comment. We carried out very detailed XPS studies on Co-TNTs in past and results are published in our earlier work by Bhattacahrya et al [26]. The hydroxide state is still preserved after 120 °C overnight annealing. To clearly reflect this in manuscript, we modified the existing sentence “We have previously shown that Co exists in Co+2 state in the form of Co(OH)2 on Co-TNTs [26]” to
“We have previously shown using detailed XPS studies that Co exists in Co+2 state (2p3/2 peak at 781.5 eV) and also Co(OH)2 is the predominant phase present on the surface of Co-TNTs [27].” (Line 177)
- Can you please describe how Co-TNT was deposited on custom-built Co-TNT packaged printed circuit board setup? Also, electrochemical sensor-oriented audience will be extremely interested in the current collector/electrodes and architecture of them as well? Please briefly provide the details.
RESPONSE: We thank the reviewer for the comment. We have incorporated the following text in the manuscript and highlighted in yellow.
The custom-made printed circuit board consists of copper clamp that holds the Co-TNT grown over the Ti sheet. The upward-facing Co-TNT side acts as a working electrode, and the bottom-facing Ti side acts as a counter electrode, to which electrical connections were made via copper lines running on top and bottom of the custom-built chip, respectively. The detailed schematic of the printed circuit was reported in our earlier work [26]. (Line 152)
- Figure 2, EDS map is not visible.
RESPONSE: We thank the reviewer for the comment. We removed the EDS map as inset from Figure 2b and made a separate Figure for better visibility (Figure 2c). Also, added EDS spectra that conforms the presence of Co (Figure 2d). The caption of Figure 2 was changed accordingly as:
Figure 2: SEM micrographs of (a) TNTs post-annealing. Inset shows sidewalls of TNTs, (b) Co-functionalized TNTs showing the Co (OH)2 precipitate, (c) EDS map of Co confirming its uniform distribution, (d) EDS spectra confirming presence of Co.
Line 175 changed to: EDS analysis confirmed the uniform distribution of Co on top of TNTs (Figure 2c), and the Co content was found to be ~ 4 wt % (Figure 2d).
- Can you please include how the S-RBD protein of SARS-CoV-2 was transferred onto the sensor surface? Is the S-RBD protein of SARS-CoV-2 in solution? What kind of solution is that? What are the properties of it?
RESPONSE: We have included the buffer composition in which S-RBD was transferred to the sensor.
The S-RBD protein in elution buffer (20mM sodium phosphate, 500mM NaCl, 8M urea, 200mM imidazole, pH 4.0) was transferred onto surface of Co-TNT using a micropipette. (Line 161)
- nano molar (nM) should have given as an explicit form in the beginning of the paper.
RESPONSE: We agree with the Reviewer and as suggested, in the beginning of the manuscript, we introduced nano molar (nM) in this revised manuscript. (Line 28)
- For the following sentence, is it known which part (elemental) of the spike protein take places/involves in that complex formation between Co and protein?
“It is hypothesized that the rapid increase in sensor response current could be attributed to the electrochemically triggered unfolding of protein that exposes its interior [27][28][29] and subsequent complex formation between Co and the protein [31][32]”
RESPONSE: We thank the reviewer for the comment. We added following discussion in the revised manuscript:
Each S-RBD protein monomeric unit contains 15 Tyrosine, 2 Tryptophan and 9 Cysteine amino acid residues [34], and all of them were reported to underdo electrochemical oxidation under application of potential [28,31]. The electrochemical oxidation process involves deprotonation, where the -OH functional group in protein is converted to -O-. We envisage that the complex formation occurs between Co+2 ion in Co-TNT and -O- radical in protein. Very similar mechanism was reported earlier, where methyl nicotinate biomarker is exposed to Co-TNT [26]. (Line 205)
- Please clarify the following sentence “(ii) Co-TNTs of even higher length.”
RESPONSE: We thank the reviewer for the comment. We mean that by use of longer Co-TNTs, which processes higher surface area, the sensor response can be increased.
- For the following sentence, the concentrations tested in the current work is similar concentration can be found in the human nasal liquids?
“Amperometry electrochemical studies indicated that the sensor could detect the protein in the concentration range 14 nM to 1400 nM.”
RESPONSE: This is one of the very important question, and based on the published data, we estimate SARS-CoV-2 viral copies and the amounts of S-RBD protein present on swab will be within the range of detection by our sensor.
This was estimated from the data published in “SARS-CoV-2 by numbers by Bar-On et al. eLife 2020;9: e57309. DOI: https://doi.org/10.7554/eLife.57309
Concentration of the virus in nasopharynx: 106-109 RNA/swab and Throat is 104-108 RNAs/swab
There are 300 copies of the spike monomeric copies/virus.
i.e. ~108-1011 copies of the Spike-RBD, which will provide detectable levels of S-RBD by our sensor.

Round 2
Reviewer 1 Report
The suggested requests have been sufficiently satisfied.